# Improvement in Quality of Life with Pelvic Floor Muscle Training and Biofeedback in Patients with Painful Bladder Syndrome/Interstitial Cystitis

**DOI:** 10.3390/jcm10040862

**Published:** 2021-02-19

**Authors:** Pedro-Santiago Borrego-Jimenez, Javier Flores-Fraile, Bárbara-Yolanda Padilla-Fernández, Sebastián Valverde-Martinez, Agustín Gómez-Prieto, Magaly Teresa Márquez-Sánchez, José-Antonio Mirón-Canelo, María-Fernanda Lorenzo-Gómez

**Affiliations:** 1Physiotherapy Department of Institute of Applied Technology, Abu Dhabi 3798, United Arab Emirates; pedro.borrego@gmail.com; 2Department of Health Sciences and Education, University UDIMA, 28400 Madrid, Spain; 3Department of Surgery, University of Salamanca, 37007 Salamanca, Spain; sebasv_2000@hotmail.com (S.V.-M.); mflorenzogo@yahoo.es (M.-F.L.-G.); 4Departamento de Cirugía, Universidad de La Laguna, 38200 Tenerife, Spain; padillaf83@hotmail.com; 5Renal Urological Multidisciplinary Research Group (GRUMUR), Institute of Biomedical Research of Salamanca (IBSAL), 37007 Salamanca, Spain; magalymarquez77@gmail.com (M.T.M.-S.); miroxx@usal.es (J.-A.M.-C.); 6Department of Urology, University Hospital of Ávila, 05004 Ávila, Spain; 7Department of Emergency, University Hospital of Salamanca, 37007 Salamanca, Spain; chicato1973@yahoo.es; 8Department of Biomedical and Diagnostic Sciences, University of Salamanca, 37007 Salamanca, Spain; 9Department of Urology, University Hospital of Salamanca, 37007 Salamanca, Spain

**Keywords:** quality of life, interstitial cystitis, inflammation, biofeedback, physiotherapy

## Abstract

Objective: To prove the benefits of pelvic floor muscle training with biofeedback (BFB) as a complementary treatment in women with bladder pain syndrome/interstitial cystitis (BPS/IC). Methods: Prospective, randomized study in 123 women with BPS/IC. Groups: BFB+ (n = 48): women with oral drug treatment (perphenazine and amitriptyline) plus intravesical instillations (sodium hyaluronate) plus pelvic floor muscle training with BFB; BFB−: (n = 75): women with oral drug treatment plus intravesical instillations. Variables: age, body mass index (BMI), time of follow-up, length of disease, time free of disease, diseases and health conditions concomitant, and responses to the SF-36 health-related quality of life questionnaire at the first consultation (SF-36 pre-treatment), and at the end of the study (SF-36 post-treatment). The treatment was considered successful when the SF-36 score reached values equal to or greater than 80 points or when the initial value increased by 30 or more points. Results: Mean age was 51.62 years old (23–82). BMI was higher in BFB−. The mean length of BPS/IC condition was 4.92 years (1–20), shorter in BFB+ than in BFB−. Mean SF-36 score pre-treatment was 45.92 points (40–58), lower in BFB+ than in BFB−. Post-treatment SF-36 score was higher than pre-treatment SF-36 score both in BFB+ and BFB−. SF-36 values were higher in BFB+ compared to BFB− over the follow-up. Conclusions: BFB improves quality of life in women with BPS/IC as adjunct therapy to combined oral and intravesical treatment.

## 1. Introduction

Although bladder pain syndrome/interstitial cystitis (BPS/IC) has been included within myofascial pain and neuropathic pain syndromes interrelated with immune and inflammatory systems [1], recent animal and clinical research indicated that many of the mechanisms for chronic pelvic pain syndromes are based within the central nervous system [2]. BPS/IC must be diagnosed on the basis of pain, pressure, or discomfort associated with the urinary bladder, accompanied by at least one other additional symptom, such as increased daily or nighttime urinary frequency, and exclusion of confusing processes that could cause symptoms. If indicated, cystoscopy with hydrodistension and bladder biopsy must be performed [3].

Multiple alternatives for IC treatment have been described, from oral treatments (such as amitriptyline isolated [4] or combined with perphenazine [5]) to intravesical treatments (including hyaluronic acid [2]) and non-pharmacological treatments (bladder training [6], pelvic floor muscle training [7,8,9,10] with or without biofeedback (BFB) [11]).

### 1.1. Electromyographic-Biofeedback as an Adjunct Therapy

BFB is the process of reintroduction in a biological system of the data obtained through the study of a phenomenon, leading to a modification of this system’s response [12]. It is a form of learning through a feedback loop. In other words, it is a process whereby electronic monitoring of a normally automatic bodily function is used to train someone to acquire voluntary control of that function [13]. The patient gains greater awareness of one or more physiological processes by a visual, auditory, or tactile signal [12].

Electromyographic biofeedback (EMG-BFB) was approved by the FDA in 1991, and it has proven to be effective without side effects or complications [14]. It is a key and indispensable technique for perineal and sphincteric reeducation. Information can be obtained faster by BFB than by a therapist, and it also helps the patient to gain awareness of her pelvic floor and learn to perform the perineal exercises faster, both for contraction and relaxation. BFB assists the correct performance of the exercises, and it is especially useful in patients with pelvic floor disorders [15,16].

### 1.2. Pelvic Floor Muscle Training for BPS/IC Treatment

Pelvic floor muscle training has shown effectiveness in the treatment of myofascial pain and the release of trigger points [9], as well as in the management of high-tone pelvic floor dysfunction [17]. Randomized clinical trials have demonstrated the greater benefit of myofascial physical therapy towards global therapeutic massage [10]. However, treatment modalities and protocols focused on BPS/IC vary between studies, and they are usually provided in combination with psychotherapeutic interventions and medical management, making it difficult to determine the ‘stand-alone’ value of physiotherapy [18]. In fact, during the Global Interstitial Cystitis/Bladder Pain Society (GIBS) Meeting 2019 in Mumbai, India [19], the importance of multimodal treatment in BPS/IC patients was stressed, and both physical and behavioral therapy are recommended in all treatment protocols besides drug treatment.

In the same line, Khandwala and Cruff performed a single-arm pilot study evaluating the effect of yoga practice in patients with BPS/IC, and they found statistically significant improvements in social function and pain components of the SF-36 [20].

The objective of biofeedback of the pelvic floor is to reestablish a balance in the physiological functioning of the pelvic structures, through regular training that breaks the stress that the various pathologies can cause in the pelvic floor, including long-term painful conditions, which will cause reflex contractures. Biofeedback acts retroactively from the reflex effector organs towards the nervous structures. It aims to regulate reflections in an inverse way. This has already been demonstrated in previous investigations by our research group [21].

### 1.3. Quality of Life and Health Indicators

Health indicators provide necessary and adequate data to evaluate the effectiveness of an intervention and its results in terms of health-related quality of life (HRQOL). They also allow professional decision-making regarding satisfaction and the overall perspective of patients [22].

The Short Form-36 Health Survey (Appendix A: Health-Related Quality of Life Questionnaire Used: SF-36) is a self-reported measure of health that is often used as a measure of a person or population’s quality of life (QoL) [23]. The SF-36 questionnaire has shown good validity, reliability, and sensitivity to changes, meeting five or more quality metrics criteria, which enabled this instrument to obtain a grade A recommendation. Different versions have shown good metric properties in diverse patients, populations, and countries. It has been proven to be an effective and reliable instrument for measuring clinical results, detecting both positive and negative changes in health [24].

### 1.4. Objective

To evaluate the effect of pelvic floor muscle training with BFB on HRQOL as a complementary treatment in patients with BPS/IC.

## 2. Method

An international, multicenter, prospective randomized study was conducted on a sample of 123 women with BPS/IC. Inclusion was randomized by each doctor prospectively between December 2013 and December 2016. The following centers participated in the study: Hospital Universitario de Salamanca (37.007 Salamanca, Spain), Hospital Universitario Virgen de la Vega (37.007 Salamanca, Spain), Hospital Universitario Virgen del Castañar (37.700 Béjar, Salamanca, Spain), Hospital Universitario Nuestra Señora de Sonsoles (05004, Ávila, Spain), and the Institute of Applied Technology (Physiotherapy Department), Abu Dhabi, United Arab Emirates.

### 2.1. Sample Selection

Patients with chronic BPS/IC were identified in the outpatient clinics for primary care, gynecology, rehabilitation, physiotherapy, and urology; all patients were referred to and diagnosed in urology.

General and pelvic floor physical examination, including neurological examination and the identification of painful trigger points, were performed in all patients. Other tests completed included a blood test (hematimetry, general biochemistry including creatinine and glomerular filtration rate); routine and urinary sediment analysis; urine culture for bacteria, fungi, and Koch bacillus; urinary cytology; abdominal ultrasound; and cystoscopy. For inclusion, signs of cystitis on cystoscopy, and inflammation findings in the bladder biopsy were needed: lesion types 2C and 3C of the European Society for the Study of Interstitial Cystitis were included: that is, the presence of glomerulations (2C) or Hunner’s lesion (3C) [3]. All patients received on paper at the first follow-up daily visit a monthly card, where the days were expressed in squares. If the patient had incidents, she recorded them at the time of suffering them. Likewise, if she was without symptoms, she also wrote it down at the end of the month. Thus, all the patients delivered the evolutionary diaries in the subsequent period. This was done in all patients in both groups. This procedure tries to avoid forgetting.

Inclusion criteria: Women 18 years or older with a diagnosis of BPS/IC with vesical inflammatory signs or symptoms (ESSIC 2C–3C).

Exclusion criteria: patients with urinary lithiasis, severe urinary incontinence, urinary infection, urological and/or pelvic malignancies, congenital abnormalities of the upper and/or lower urinary tract, neurogenic bladder, intermittent catheterization, indwelling catheter; pregnancy; patients unable to give informed consent to participate in the study.

### 2.2. Study Groups

BFB+ (*n* = 48): patients who received oral drug treatment plus intravesical instillations and additional treatment with pelvic floor muscle training with biofeedback of the pelvic floor with electromyography (BFB-EMG). BFB− (*n* = 75): patients who received oral drug treatment plus intravesical instillations without additional BFB-EMG.

Oral drug treatment was a combination of perphenazine (2 mg) plus amitriptyline (25 mg) once daily as a continuous treatment.

Intravesical treatment consisted of instillations of sodium hyaluronate (40 mg) (one intravesical instillation per week for 4 weeks).

Patients were randomly allocated to either group by each investigator until BFB+ reached 48 patients and BFB− had 75 patients. The total number of patients included in each group was decided on depending on the treatment capacity in each modality; treatment modalities were subject to the number of schedule slots in the consultation and treatment cabinets.

### 2.3. Procedure

The BFB-EMG therapeutic program consisted of physiotherapist-guided therapeutic sessions where the patient handled a signal on a screen using her pelvic floor muscles. Each session lasted 20 min, and they were scheduled once a week for 20 weeks. The patient lay flat in the supine position, with a slight hip flexion and lumbar lordosis protection to avoid fatigue. In this position, the patient was able to see the screen of the BFB-EMG device with the scene. Pre-gelled pediatric auto-adhesive electrodes were used.

After a brief explanation of pelvic floor anatomy by the physiotherapist, the patient was trained to contract the pelvic floor muscles for 3–5 s and relax for 6–8 s. These contractions were recorded, reflecting muscle tone and power, as well as the duration of the entire perineal registration. Each signal was continuously registered on a polygraph.

### 2.4. Studied Variables

Age, body mass index (BMI), length of follow-up from the first consultation until the last follow-up (in days), length of disease (in years), concurrent conditions or diseases, concurrent treatments, disease-free time, responses to the SF-36 HRQOL questionnaire Spanish Version (SF-36-HRQOL) [19] at the first consultation and 3, 6, and 12 months after treatment. The same diagnostic protocol was followed in all patients, and the same schedule of visits was established: first consultation for inclusion in the study, and at 3, 6, and 12 months after concluding the treatment.

“SF-36 pre-treatment” was defined as the SF-36-HRQOL score at the first consultation when the BFB-EMG was indicated. “SF-36 post-treatment” was the mean value of the results obtained in the questionnaire at 3, 6, and 12 months after concluding the treatment.

SF-36-HRQOL was interpreted as the impairment in quality of life related with a specific health problem, in a range between 0 and 100, with 0 being the worst possible status and 100 being complete health and the absence of any discomfort caused by the health problem.

Disease-free time was defined as the length (in days) that patients reported to be completely asymptomatic without any treatment during the follow-up.

The result was considered successful in the following cases:
Result of SF-36-HRQOL questionnaire equal or greater than 80 points; orIncrease in 30 or more points in the SF-36-HRQOL questionnaire compared to the initial score.

The result was considered a failure in the following cases:Result of SF-36-HRQOL questionnaire below 80 points; andIncrease lower than 30 points in the SF-36-HRQOL questionnaire compared to the initial score.

### 2.5. Statistical Analysis

The analysis was performed using the NSSS2006/GESS2007 statistical system (NCSS, LLC; Kaysville, UT, USA). Descriptive statistics, Student’s t-test, chi-square test, Fisher’s exact test, ANOVA (Scheffé’s test for normal samples and Kruskal–Wallis for other distributions), and multivariate analysis. Statistical significance was accepted for *p* < 0.05.

### 2.6. Ethical Issues

The study protocol with code 230/284/1/2 was approved by the Drug Research Ethics Committee of the Healthcare Area of Ávila, Spain.

## 3. Results

### 3.1. Descriptive Statistics

The mean age of the whole sample was 51.62 years (SD 1.92, median 55, range 23–82). Average BMI was 24.92 kg/m^2^ (SD 0.53, median 24.08, range 18.75–35.16). Except for two patients in the BFB+ failure group who did not carry out the control at 12 months post-treatment, the rest of the patients completed all the controls.

The mean value for the total time of follow-up from the first consultation until the last follow-up (completion of the study) was 2342.02 days (SD 13.08, median 2150, range 365–4380). The mean number of years after the diagnosis of BPS/IC prior to starting the treatment (YPT) was 4.92 years (SD 0.39, median 3.2, range 1–20), with a longer follow-up in patients in BFB− (Table 1).

In the BFB+ group, there was more hypothyroidism, more than two concomitant diseases, more concomitant treatments with benzodiazepines, topical oestrogens, third-step analgesics, endovesical treatments with glycosaminoglycans, toxic habits, and history of dystocic deliveries, compared to the BFB− group; while in the BFB− group there were more concomitant treatments with anticholinergics, a history of hysterectomy and euthotic deliveries (Table 1).

### 3.2. Results in Quality of Life

In BFB+, a greater proportion of patients underwent successful treatment (n = 36, 75.00%) than in BFB− (n = 44, 58.67%). Baseline scores were similar in both groups. There were no differences between SF-36 score pre-treatment in successful and failed patients in both BFB+ (*p* = 0.7799) and BFB− (*p* = 0.4331). Logically, responses to SF-36 post-treatment were higher in successful than in failed patients in both groups (BFB+, *p* = 0.0001; BFB− *p* = 0.0001) (Table 2).

The patients received the same follow-up in the two groups. The patients similarly complied with the prescribed treatment regimen in both groups, with the exception of the loss to follow-up of two patients in BFB+, who were included in the failure outcome. Therefore, two patients were lost in the multivariate analysis in BFB+. They were treated in the Hospital Universitario Nuestra Señora de Sonsoles (05004, Ávila, Spain): BFB+ success: *n* = 3; BFB+ failure: *n* = 1; BFB− success: *n* = 3; BFB− failure: *n* = 1; in the Institute of Applied Technology (Physiotherapy Department), Abu Dhabi. United Arab Emirates: BFB+ success: *n* = 3; BFB+ failure: *n* = 1; BFB− success: *n* = 3; BFB− failure: *n* = 1; in the Hospital Universitario de Salamanca (37.007 Salamanca, Spain): BFB+ success: *n* = 11; BFB+ failure: *n* = 3; BFB− success: *n* = 14; BFB− failure: *n* = 10; in the Hospital Universitario Virgen de la Vega (37.007 Salamanca, Spain): BFB+ success: *n* = 11; BFB+ failure: *n* = 3; BFB− success: *n* = 11; BFB− failure: *n* = 10; and in the Hospital Universitario Virgen del Castañar (37.700 Béjar, Salamanca, Spain): BFB+ success: *n* = 8; BFB+ failure: *n* = 4, but 2 were lost to follow-up; BFB− success: *n* = 13; BFB− failure: *n* = 9.

A logistic regression analysis was performed to find out the relative risk presented by a patient presenting the analyzed variables of being treated with adjuvant biofeedback, that is, of being included in the BFB+ group or in the BFB− group, identifying the relative risk with a confidence interval of 95 %, 76%, and 58%.

An omnibus test was previously carried out to determine the relationship of the variables with the indication of adjuvant treatment with BFB, finding statistical significance (*p* = 0.0078): R2 = 87.177. In other words, it was a powerful statistical analysis, and it was found that the proportion of the variance explained in the regression model and the classification of the individuals in the BFB+ and BFB− groups is correct and the analysis adequate.

In the RR analysis with a 95% confidence interval, it was found that women with the following variables were more likely to be in the BFB+ group: older (*p* = 0.001), higher BMI (*p* = 0.016), lower SF36-pretreatment (gradient of the equation −0.232) (*p* = 0.0029), longer evolution time in years (*p* = 0.016), other pathological conditions or concomitant diseases (*p* = 0.010), other pharmacological treatments (*p* = 0.015), active smoker (*p* = 0.025), and more than two concomitant treatments (*p* = 0.0014).

In the RR analysis with a confidence interval of 76% and 58%, the same distribution was found, except for the high BMI, which was not higher (*p* = 0.828).

Likewise, a greater probability of a higher value of SF-36 is observed in patients receiving adjuvant treatment with BFB both in the 95% confidence interval (RR 1.026; *p* = 0.049), and in the interval of the 76% confidence interval (RR 1.026; *p* = 0.049) or in the 58% interval (RR 1.026; *p* = 0.049) (Table 3).

In the general sample, a correlation between the SF36-post-treatment and the following variables had a negative correlation—that is, the lower the value of the variable, the higher and better the quality of life was measured by the SF-36 questionnaire after treatment: BMI (regression coefficient −0.188, *p* = 0.006); diabetes mellitus (DM) (regression coefficient −5.142, *p* = 0.0.038), “other pathological conditions” (regression coefficient −1.444, *p* = 0.025); “other pharmacological treatments (regression coefficient −2.104, *p* = 0.038).

The correlation of the SF36-post-treatment and the following variables had a positive correlation—that is, the higher the value of the variable, the higher and better the quality of life was measured by the SF-36 questionnaire after treatment: overactive bladder (regression coefficient 3357, *p* = 0.023), intravesical instillation of glycosaminoglycans (regression coefficient 2.113, *p* = 0.028).

In the BFB+ group, a negative correlation was observed between SF36-post-treatment and the following variables: BMI (regression coefficient −1.013, *p* = 0.00019), the SF-36 pretreatment (regression coefficient −0.777, *p* = 0.003), and evolution time or time of suffering from cystitis in years (regression coefficient −1.325, *p* = 0.000085).

The multiple regression (Table 4) analysis in the BFB− group found a negative correlation between the SF36-post-treatment and the following variables: “other pathological conditions” (regression coefficient −1.839, *p* = 0.0003) and “other pharmacological treatments (regression coefficient −2.264, *p* = 0.057). There was a positive correlation between the SF36-post-treatment value and overactive bladder (regression coefficient 3559, *p* = 0.007, treatment with benzodiazepines (regression coefficient 2038, *p* = 0.062), and intravesical instillation of glycosaminoglycans (regression coefficient 2870, *p* = 0.008).

In the regression models, variables that have a high correlation with the dependent variable were excluded, since they present high variability, such as arterial hypertension, depression, overactive bladder, urinary incontinence, non-repair cystocele, urinary incontinence surgery, toxic habit plus surgical interventions, caesarean section, more that two concurrent medical conditions, urinary tract infections, mild or level 1 urinary incontinence, anticholinergics, topical estrogens, third-step analgesic drugs, amitriptyline antidepressant, more than two concurrent treatments, smoker, toxic habit plus concurrent medical conditions, repaired cystocele, curettage, more than two concurrent surgical interventions, and dystocic delivery.

Figure 1 shows the plot for multiple regression, and thus shows the relationship of the variables investigated with the dependent variable—that is, with the result of the improvement in quality of life measured with the SF-36 questionnaire, after the treatments received in the general sample.

## 4. Discussion

Patients with BPS/IC experience great suffering, and progress in disease understanding has been made in the last few decades. In the absence of well-defined mechanisms, describing the condition by its symptoms, signs and, where possible, by investigations has been demonstrated to have clinical and research validity in many situations [2]. Therefore, chronic pelvic pain phenotyping is of paramount importance in the diagnosis, follow-up, and treatment schedule of this syndrome. There is no standardized therapeutic management, but rather a sequential treatment protocol must be agreed upon for each patient, to test treatments and change them according to the patient’s response. In addition, multimodal protocols can be offered so that complementary measures improve the effectiveness of each one applied individually.

In the management of BPS/IC in our setting, pelvic floor muscle training with BFB and surface electrodes is indicated in many patients as adjunctive treatment [11]. Surface EMG is a non-invasive contact method that is used by placing surface electrodes on the skin. It is commonly used instead of needle EMG, but some authors criticize it, saying that the surface registry can be affected by the activity of other muscles and that the amplitude of the registry is very low [25]. However, needle EMG is more invasive and uncomfortable, and it also has another disadvantage: the needle acts not only in the motor units of the muscle (or muscle group) intended to be examined, but also in unmyelinated fibers that collect the pain caused by the puncture. This painful stimulus can trigger local reflexes, affecting the registry [26]. To assess the efficacy of periurethral sphincter/pelvic floor surface EMG, a study in children with voiding disorders was conducted [27]. Correlation between both groups had a high statistical significance: there was no difference between the results collected by the perianal surface EMG and the urethral surface EMG.

We wanted to thoroughly analyze patients’ medical background, focusing on concomitant disorders and other chronic drug treatments in order to avoid biases and interferences in the results of the BPS/IC treatments. The medical and surgical backgrounds of those patients who improved their SF-36 score more than 30 points after treatment were studied. Although the statistical significance could be influenced by the fact that there were less than five patients in BFB−, there is a higher proportion of patients who underwent surgical correction for UI and cystocele in BFB+, as well as for two or more surgical interventions. In addition, patients in BFB+ had more dystocic deliveries and they took more frequently benzodiazepines, topical estrogens, and third-level analgesics. It has been previously published that a history of dystocic deliveries has been associated with a worse prognosis for pelvic floor treatments [28], and this factor could favor a worse HRQOL in patients in BFB+. However, they also had a need for treatment with benzodiazepines and third-level analgesics, which could reinforce the benefit provided by pelvic floor muscle training with BFB, balancing the baseline health status of the patients in both groups.

In those patients with insufficient relief after treatment, it is observed that those women in BFB+ have a high rate of concomitant treatment with benzodiazepines (83.33%) and third-level analgesics (66.66 %), which coincides with higher rates of depression (50%) and other diverse pathological conditions (91.66%). They also have a high occurrence of more than two concomitant diseases, including women who have concomitant toxic habits (41.66%). Therefore, in women in whom the disease did not improve and who received concomitant BFB, the baseline health status of the patients was lower than that in patients who did not receive BFB.

The SF-36 questionnaire is a generic questionnaire on HRQOL, used internationally and validated in Spanish [24]. The great utility of this questionnaire as an indicator of HRQOL has been specifically proven in pelvic floor disorders [29,30]. Patients in BFB+ showed a lower quality of life score at the beginning of the study. The improvement of patients in BFB+ was much higher (75%) compared to the BFB− group that did not received BFB (58.66%). This difference is significant, as proved by the multivariate analysis.

According to the results of our study, BFB as an adjunct treatment has a great contribution to the improvement of BPS/IC patients, who are principally patients who have a great affliction and a complex management of associated painful symptoms.

Therefore, this study, which investigates the efficacy of pelvic floor muscle training with BFB as an adjuvant treatment, is an important contribution to the management of BPS/IC in routine medical practice. Furthermore, the application of this treatment modality in women with worse health status has achieved good results in disease control. Hence, the results obtained support the implementation of pelvic floor muscle training with BFB as a useful measure in patients with BPS/IC, especially as no side effects have been found with the application of BFB.

The limitations of this study are those inherent to the interstitial cystitis disease itself. All patients were diagnosed following the same diagnostic protocol. The two main types of interstitial cystitis were taken into account, as specified in Methods: the definition of lesions in the cystoscopy of the European Society for the Study of Interstitial Cystitis was used for the study of C2 and C3 interstitial cystitis [3]. Despite this, it is known that there is variability in the disease between different patients, since it is difficult in this disease to identify precise stages.

On the other hand, a very strong point is the long experience of the research team in the management of the disease and the coherence and rigor of the follow-up protocols in all patients [1,11,31,32,33].

## 5. Conclusions

Pelvic floor muscle training with BFB increases the probability of improving quality of life in patients with BPS/IC. It provides significant benefit when used as an adjunct therapy to oral and intravesical combined baseline treatment, even in patients with worse health-related quality of life conditions.

## Figures and Tables

**Figure 1 jcm-10-00862-f001:**
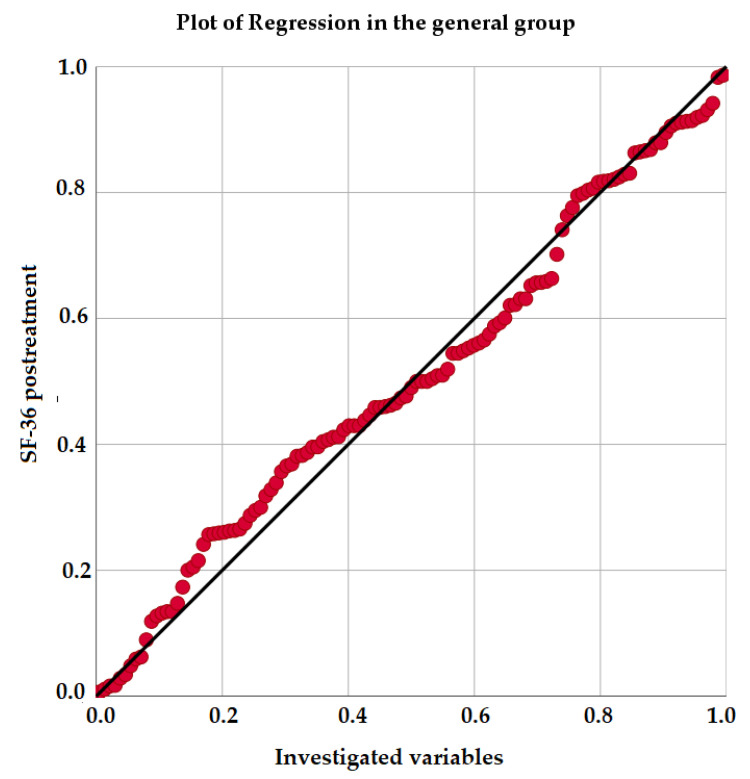
Plot for logistic regression in the general sample.

**Table 1 jcm-10-00862-t001:** Characteristics of the patients in BFB+ and in BFB−.

	BFB+				BFB−				*p*
	Mean	SD	Median	Range	Mean	SD	Median	Range	
Age (y.o)	46.12	2.04	49	24–77	56.82	1.34	58	23–82	0.0535
BMI	24.16	0.64	21.78	19.63–35.16	25.47	0.35	25.95	18.75–29.97	0.04463
TFU	2321.05	16.47	2200	1650–4380	2392.97	10.56	2098	1247–5720	0.8502
YPT	3.2	0.17	2.5	1–6	6.58	0.49	5	1–20	0.0391
SF-36 basal	42.98	2.44	42.00	41–47	47.96	6.17	45.00	40–58	0.0046
Concurrent diseases or conditions									
					BFB− (*n* = 75) *n* = 48	BFB+ (*n* = 48) *n* = 75	*p*
					n	%		n	%
HT					0	0.00	5	6.67	0.1553
DM					0	0.00	3	4.00	0.2803
Hypothyroidism					10	20.83	0	0.00	0.0001
Depression					6	12.50	6	8.00	0.5355
Other pathological conditions or diseases					32	66.67	45	60.00	0.5677
More than 2 concurrent medical conditions					6	12.50	1	1.33	0.0140
History of UTI					0	0.00	1	1.33	1.0000
Overactive bladder					5	10.42	2	2.67	0.1088
UI					0	0.00	1	1.33	1.0000
Mild or level 1 urinary incontinence					0	0.00	4	5.33	0.1553
Non-repaired cystocele					0	0.00	2	2.67	0.5202
Non-repaired rectocele					0	0.00	1	1.33	1.0000
Concurrent treatment with benzodiazepine					19	39.58	9	12.00	0.0007
Concurrent treatment with anticholinergics					0	0.00	9	12.00	0.0118
Concurrent treatment with topical oestrogens					10	20.83	3	4.00	0.0052
Concurrent treatment with second-step analgesic drugs					0	0.00	3	4.00	0.2803
Concurrent treatment with third-step analgesic drugs					19	39.58	11	14.67	0.0024
Concurrent treatment with amitriptyline antidepressant					14	29.17	15	20.00	0.2797
Treatment with intravesical instillation of GAG					26	54.17	25	33.33	0.0256
Other pharmacological treatments					24	50.00	27	36.00	0.1373
More than 2 concurrent treatments					19	39.58	26	34.67	0.7014
Smoker					5	10.42	10	13.33	0.7802
Toxic habit plus concurrent medical conditions					5	10.42	0	0.00	0.0079
Surgical Background									
UI surgery					10	20.83	8	10.67	0.1898
Repaired cystocele					6	12.50	8	10.67	0.7771
Curettage					0	0.00	2	2.67	0.5202
Hysterectomy					0	0.00	7	9.33	0.0419
Other surgical interventions					17	35.42	17	22.67	0.1495
Gynecological cancer surgery					0	0.00	1	1.33	1.0000
History of more than 2 concurrent surgical interventions					11	22.92	11	14.67	0.3348
Toxic habit plus history of concurrent surgical interventions					0	0.00	2	2.67	0.5202
Obstetric Background									
Cesarean section					0	0.00	2	2.67	0.5202
Eutocic delivery					0	0.00	12	16.00	0.0032
Dystocic delivery					10	20.83	1	1.33	0.0003

BMI: body mass index; BFB+: group of patients who received adjuvant biofeedback (BFB) of the pelvic floor; BFB−: group of patients who did not received biofeedback of the pelvic floor; TFU: total time of follow-up; y.o.: years old; YPT: years of BPS/IC prior to starting the treatment.

**Table 2 jcm-10-00862-t002:** SF-36 scores in BFB+ and BFB−.

	SF-36					
Group	Check Point	Media	SD	Median	Range	*p*
BFB+	Pre-treatment	42.98	2.44	42	41–47	
	Post-treatment 3 months	78.27	13.02	85	51–92	0.0092
	Post-treatment 6 months	78.31	13.40	85	49–92	
	Post-treatment 12 months	75.72	12.81	81	47–90	
	Post-treatment average	77.29	13.20	84	51.89	
BFB+ success	Pre-treatment	42.81	2.66	41	41–47	
	Post-treatment 3 months	85.38	3.51	86	77–92	0.2284
	Post-treatment 6 months	85.52	4.16	86	76–92	
	Post-treatment 12 months	82.08	3.84	81	73–90	
	Post-treatment average	84.55	3.15	85	78–89	
BFB+ failure	Pre-treatment	43.50	1.57	43.5	42–45	
	Post-treatment 3 months	56.92	4.68	56.5	51–62	0.8499
	Post-treatment 6 months	56.67	5.50	57	49–62	
	Post-treatment 12 months	52.80	4.21	53.5	47–58	
	Post-treatment average	55.50	4.7	55.50	51–60	
BFB−	Pre-treatment	47.96	6.17	45	40–58	
	Post-treatment 3 months	73.92	15.97	83	48.95	0.0375
	Post-treatment 6 months	72.25	15.69	81	48.90	
	Post-treatment 12 months	69.76	16.38	80	46–87	
	Post-treatment average	71.79	15.79	82	50–82	
BFB− success	Pre-treatment	48.20	5.78	45	40–58	
	Post-treatment 3 months	86.90	3.58	87	80–95	0.1028
	Post-treatment 6 months	85	3.23	85	78–90	
	Post-treatment 12 months	83.27	2.43	83	78–87	
	Post-treatment average	84.77	2.44	85	80–88	
BFB− failure	Pre-treatment	47.61	6.75	45	41–58	
	Post-treatment 3 months	55.48	3.45	55	48–61	0.2185
	Post-treatment 6 months	54.16	3.91	53	48–61	
	Post-treatment 12 months	50.58	2.40	50	46–55	
	Post-treatment average	53.35	2.85	52	50–59	

SF-36: value in the answers to the quality of life questionnaire SF-36. SD: standard deviation. BFB+: group of patients who received adjuvant biofeedback of the pelvic floor; BFB−: group of patients who did not receive biofeedback of the pelvic floor.

**Table 3 jcm-10-00862-t003:** Logistic regression analysis of the relative risk of women who have the analyzed variables of being treated with adjuvant BFB.

Variables	RR	*p* Value	95% C.I.		RR	*p* Value	76% C.I.		RR	*p* Value	58% C.I.	
			Lower	Upper			Lower	Upper			Lower	Upper
Age	0.958	0.001	0.935	0.982	0.958	0.001	0.944	0.972	0.958	0.001	0.948	0.968
BMI	0.640	0.016	0.910	1.078	0.991	0.828	0.942	1.042	0.991	0.828	0.957	1.026
SF-36 pretreatment	0.793	0.0029	0.711	0.884	0.793	0.029	0.743	0.846	0.793	0.029	0.758	0.829
Evolution time in years	0.808	0.0011	0.711	0.918	0.808	0.001	0.748	0.872	0.808	0.001	0.766	0.852
Other pathological	0.217.	0.010	0.090	0.524	0.217	0.001	0.128	0.368	0.217	0.001	0.151	0.312
Other pharmacological treatments	0.398	0.015	0.189	0.840	1.083	0.015	0.254	0.623	0.398	0.015	0.293	0.541
Smoker	0.271	0.025	0.086	0.852	2.000	0.025	0.137	0.539	0.271	0.025	0.170	0.435
Repaired cystocele	0.330	0.046	0.111	0.980	0.330	0.046	0.172	0.634	0.330	0.046	0.211	0.517
Dystocic delivery	7.231	0.064	0.895	58.437	7.231	0.064	2.066	25.306	7.231	0.064	3.061	17.083
More 2 concurrent medical conditions	4.087	0.199	0.476	35.058	4.087	0.199	1.127	14.824	4.087	0.199	1.688	9.896
Overactive bladder	1.643	0.563	0.306	8.828	1.643	0.563	0.600	4.502	1.643	0.563	0.822	3.281
Benzodiazepine	1.576	0.315	0.649	3.827	1.576	0.315	0.926	2.682	1.576	0.315	1.094	2.270
Topical oestrogens	2.308	0.223	0.601	8.858	2.308	0.223	1.030	5.169	2.308	0.223	1.327	4.013
Amitriptyline antidepressant	0.505	0.112	0.217	1.173	0.958	0.001	0.944	0.972	0.958	0.001	0.948	0.968
Intravesical instillation of GAG	0.488	0.057	0.233	1.023	0.991	0.828	0.942	1.042	0.991	0.828	0.957	1.026
More than 2 concurrent treatments	0.287	0.0014	0.133	0.620	0.793	0.029	0.743	0.846	0.793	0.029	0.758	0.829
Other surgical interventions	0.495	0.088	0.220	1.111	0.808	0.001	0.748	0.872	0.808	0.001	0.766	0.852
More than 2 concurrent surgical interventions	0.578	0.247	0.228	1.463	0.217	0.001	0.128	0.368	0.217	0.001	0.151	0.312
SF-36 post-treatment	1.026	0.049	1000	1053	1.026	0.049	1010	1042	1.026	0.049	1.015	1.037

BFB: biofeedback of pelvic floor. RR: relative risk. CI: confidence Interval. BMI: body mass index; SF-36: value in the answers to the quality of life questionnaire SF-36. GAG: glycosaminoglycans.

**Table 4 jcm-10-00862-t004:** Multiple regression analysis’ table: probability of quality of life’s improvement or worsening: relationship between the result in the SF-36 questionnaire post-treatment and the variables.

Multiple Regression Analysis in the General Sample					
	Unstandardized Coefficient	Standardized Coefficients	*p* Value	95.0% Confidence Interval for B	
				Lower Bound	Upper Bound
Age	−0.001	−0.001	0.969	−0.034	0.033
BMI	−0.188	−0.053	0.006	−0.320	−0.056
SF-36 pretreatment	0.065	0.024	0.247	−0.045	0.175
Evolution time in years	−0.020	−0.006	0.751	−0.144	0.105
Time disease free	−0.012	−0.008	0.712	−0.074	0.050
DM	−5.142	−0.054	0.038	−9.990	−0.294
Other pathological	−1.444	−0.047	0.025	−2.705	−0.182
Overactive bladder	3.357	0.053	0.023	0.464	6.250
Non-repaired rectocele	3.035	0.018	0.423	−4.447	10.518
Benzodiazepine	0.619	0.018	0.520	−1.282	2.520
Intravesical instillation of GAG	2.113	0.070	0.028	0.227	3.998
Other pharmacological treatments	−2.104	−0.069	0.038	−4.085	−0.123
Hysterectomy	−1.272	−0.013	0.670	−7.174	4.629
Other surgical interventions	1.124	0.034	0.330	−1.156	3.404
Gynecological cancer surgery	0.869	0.005	0.767	−4.925	6.663
Cesarean section	2.253	0.019	0.546	−5.118	9.624
Eutocic delivery	−0.930	−0.019	0.492	−3.605	1.745
Results	−30.580	−0.976	0.0005	−32.020	−29.139
**Multiple Regression Analysis in BFB+**					
	**Unstandardized Coefficient**	**Standardized Coefficients**	***p*-Value**	**95.0% Confidence Interval for B**	
				Lower Bound	Upper Bound
Age	−0.042	−0.062	0.186	−0.106	0.022
BMI	−1.013	−0.374	0.00019	−1.227	−0.798
SF-36 pretreatment	−0.777	−0.148	0.003	−1.272	−0.282
Evolution time in years	−1.325	−0.180	0.000085	−1.916	−0.733
Time disease free	−0.008	−0.003	0.926	−0.173	0.157
DM	−0.896	−0.017	0.704	−5.677	3.885
Other pathological	−0.623	−0.019	0.555	−2.761	1.515
Overactive bladder	1.312	0.021	0.720	−6.098	8.723
Non-repaired rectocele	1.139	0.013	0.752	−6.181	8.460
Benzodiazepine	−0.475	−0.015	0.821	−4.728	3.778
Intravesical instillation of GAG	1.736	0.068	0.208	−1.022	4.493
Other pharmacological treatments	−1.645	−0.064	0.322	−4.991	1.701
Hysterectomy	0.836	0.016	0.704	−3.623	5.294
Other surgical interventions	−0.168	−0.006	0.918	−3.490	3.155
Caesarean section	0.627	0.010	0.856	−6.372	7.625
Eutocic delivery	−0.859	−0.030	0.482	−3.328	1.610
Results	22.524	0.755	0.00095	26.690	18.359
**Multiple Regression Analysis in BFB−**					
	**Unstandardized Coefficient**	**Standardized Coefficients**	***p*-Value**	**95.0% Confidence Interval for B**	
				Lower Bound	Upper Bound
Age	−0.016	−0.013	0.452	−0.058	0.026
BMI	0.011	0.003	0.871	−0.128	0.151
SF-36 pretreatment	0.047	0.018	0.373	−0.058	0.152
Evolution time in years	0.0001	0.0003	0.998	−0.105	0.104
Time disease free	−0.014	−0.005	0.774	−0.108	0.081
Other pathological	−1.839	−0.059	0.003	−3.020	−0.659
Overactive bladder	3.559	0.057	0.007	1.002	6.116
Benzodiazepine	2.038	0.057	0.062	−0.107	4.182
Intravesical instillation of GAG	2.870	0.087	0.008	0.780	4.961
Other pharmacological treatments	−2.264	−0.067	0.057	−4.594	0.066
Other surgical interventions	−0.443	−0.012	0.739	−3.088	2.203
Gynecological cancer surgery	0.960	0.007	0.684	−3.738	5.658
Results	−31.568	−0.991	0.0005	−32.842	−30.293

BMI: body mass index; BFB+: group of patients who received adjuvant biofeedback of the pelvic floor; BFB−: group of patients who did not receive biofeedback of the pelvic floor. SF-36: value in the answers to the quality of life questionnaire SF-36. DM: diabetes mellitus. GAG: glycosaminoglycans. Evolution time: time of suffering from interstitial cystitis.

## Data Availability

The data presented in this study are available on request from the corresponding author. The data are not publicly available due to, they will be used for further studies in progression.

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
