# Peer review of "Improvement in Quality of Life with Pelvic Floor Muscle Training and Biofeedback in Patients with Painful Bladder Syndrome/Interstitial Cystitis"

_jcm, 2021, doi:10.3390/jcm10040862_

Round 1

Reviewer 1 Report

This manuscript addressed the interesting and highly relevant question of evaluating a therapy (biofeedback) for IC/BPS, a condition that is widespread, highly debilitating, and to date has no established effective treatment.  Although the problem is interesting, there are many problems with the methods and analysis.  The design is also not clear.  The analytic approaches aren’t well described or linked to specific study goals. It’s not clear that the methods used are appropriate; there’s a running problem throughout of conditioning/stratifying on posttreatment variables (SF-36 or responder status) inappropriately. As a result, the conclusions are not well-supported, as they are based on in appropriate methodology. 

The authors have good data to address an interesting problem, but in order to use it to try draw conclusions about biofeedback efficacy they need to redo all of the analyses and then better describe their approach.  More details are below.

  • Was this a study of randomized treatments or an observational study? P.3 line 13 says observational, but p. 3 like 134 says that patients were ‘randomly’ allocated.
  • Further, what was the study schedule for interacting with patients? How frequently were patients scene, and what was the schedule for collecting different variables?
  • Do concomitant treatments differ between the group that received biofeedback and the group that did not? This should be described and compared in the GA and GB groups with all 48 and 75 participants, not separately for responders and non-responders.
  • Similarly, did patients follow the prescribed regimens equally well in both groups? This should be reported in the results.
  • The authors describe capacity limitations as justification of the imbalanced sample sizes in each arm. How did this vary by study site?
  • How was disease-free time reported? Is this reported by the patient directly to the research team, or does it come by way of the physician during an in-clinic visit?  Related, what are the possible values that it can take?  If at an office visit, are there prespecified times at which it can be reported? Is there any possibility of ascertainment bias between the two groups when measuring this?
  • The definitions of success and failure or not exclusive. Someone who goes from 40 to 70 satisfies both criteria, as does someone who goes from 75 to 85. The authors need an ‘or’ statement to define success, and an ‘and’ statement to define failure. If this interpretation is correct, the authors can simply state that a success is defined as increase of 30 or more points from
  • The statistical analysis section is terribly incomplete. The authors need to list WHY each type of analysis was done in the context of the study goals. They should also describe a bit more of the detail about how each of the analyses were done. For the multivariate analysis, what were the contributing variables? What was the goal of this analysis? Which analysis was done to examine the treatment effect of biofeedback? Which variables did this analysis adjust for? There’s a lot of output presented without any rationale or explanation.
  • Given the longitudinal at 3, 6, and 12 months, why not use an approach such as mixed effects models or GEE that would allow outcomes to be compared using each time points separately? The authors are losing information about the differences in trends by taking the mean over outcomes within individual. It would be of interest to see plotted the proportion of responders in each arm at each of the 3 follow up time points and then to report on whether the differences between arms are constant in time or increasing or decreasing.
  • The baseline characteristics table (Table 1) should include baseline SF36 and concurrent conditions. These should be reported in each group without consideration of responder status.
  • What was the completeness of data in each arm? How many participants provided at 1, 2, or 3 of the assessments at 3, 6, and 12 months?
  • It looks like the authors did a within group (GA and GB) analysis comparing those who were successes and those who were failures to infer treatment effects. This is not the right analysis to determine the effect of BFB, and as such, table 2 should be removed.  The authors need to do a between group analysis that compares the probability of success among those who were in the GA group vs those who were in the GB group.  Unadjusted and adjusted analyses should be presented.  It looks like n(%) responding is reported in lines 210-211 of p. 5, but there is no p-value to assess whether 75% vs 58% is statistically significant.  The authors should further report a measure of effect such as a risk difference, risk ratio, or odds ratio with respective 95% confidence interval.    The primary analysis, however, should be a between groups comparison adjusted for potential confounders, necessarily including baseline SF-36. This can be accomplished through an appropriate regression model.  A measure of effect that considers the difference in % success, adjusted for potential confounders (by definition pre-treatment), along with respective 95% CI and p-value, should be reported as the primary analysis, and differences in mean change from initial SF36, treating response as a continuous variable, should be reported as secondary. The analysis presented in table 4 is close, but it should contain treatment (BFB or not ie GA or GB) as effect, with the treatment difference determined by the coefficient of GA or GB, and post-treatment SF-36 and any other post-treatment variables (time disease free?) should not be in the model.
  • More on table 4. The logistic model appears very poorly specified. The Wald statistics and standard errors are implausibly huge. The intercept (constant) is nonsensical. This is likely b/c post-treatment SF-36 is in the model and by definition constructs the outcome. 
  • The multivariate clustering analysis should be removed. The purpose of this analysis is not stated, and the authors in appropriately combine pre- and post-treatment variables to determine factors that are predictive of success. The results are not clearly presented, and it’s not clear how the authors intend for the analysis to be used. Are they trying to determine a group likely to respond to treatment? Are they trying to assess mechanism?  Moreover, it does not make sense to include post-treatment SF-36 in the clustering since it defines the outcome. This is acknowledged in the discussion, but the authors’ justification does not hold.
  • Figures 1 and 3 are unintelligible. Why are upper and lower Cis presented but not point estimates? And why are there lines connecting points that represent different variables. These are not scatter plots as they are labeled.
  • How is ‘influence’ defined in Figure 2?
  • 10 lines 321-331 contain conclusions based on inappropriate analyses. These statements are not supported. The prognostic value of initial SF36 cannot be inferred when post-treatment SF36 is included since the impact of initial SF36 is likely mediated by posttreatment SF36. The multivariate analysis does not provide evidence of the significance of the difference in proportion of responders.  This should be done by logistic regression or Fisher’s exact test assuming 1 response per participant obtained by average longitudinal response.  The generalized linear mixed model approach mentioned above is an alternative approach that can accommodate longitudinal responses for each participant.

Additional issues:

  1. The authors statement of their goal sounds biased in the direction of a positive result. They should state that they want to ‘evaluate’ or ‘determine’ the effect of biofeedback, not to ‘prove’ it.
  2. The group names GA and GB are not intuitive. Why not label the groups as something that reflects the therapy, like BFB+ and BFB-?
  3. On p. 3 line 96 it should be ‘BRB on HRQOL’ rather than ‘BRB in HRQOL’.

Author Response

REVISOR 1:

This manuscript addressed the interesting and highly relevant question of evaluating a therapy (biofeedback) for IC/BPS, a condition that is widespread, highly debilitating, and to date has no established effective treatment.  Although the problem is interesting, there are many problems with the methods and analysis.  The design is also not clear.  The analytic approaches aren’t well described or linked to specific study goals. It’s not clear that the methods used are appropriate; there’s a running problem throughout of conditioning/stratifying on posttreatment variables (SF-36 or responder status) inappropriately. As a result, the conclusions are not well-supported, as they are based on in appropriate methodology. 

The authors have good data to address an interesting problem, but in order to use it to try draw conclusions about biofeedback efficacy they need to redo all of the analyses and then better describe their approach.  More details are below.

1.-Was this a study of randomized treatments or an observational study? P.3 line 13 says observational, but p. 3 like 134 says that patients were ‘randomly’ allocated.

The reviewer is right: there is a misprint on line 99: we correct: the inclusion of patients in the study was randomized.

We change it in the abstract also.

2.-Further, what was the study schedule for interacting with patients? How frequently were patients scene, and what was the schedule for collecting different variables?

As expressed in lines 108-110 and lines 152-156, the patients were diagnosed with interstitial cystopathy and were known to various specialists (lines 108-110) or were suspected of having interstitial cystopathy. All these patients were referred to the Urology consultation to offer them to participate in the study.As expressed in lines 152-156, the same diagnostic protocol was followed in all patients and the same schedule of visits was established: first consultation for inclusion in the study, at 3, 6 and 12 months after concluding the treatment.We add this paragraph.  

3.-Do concomitant treatments differ between the group that received biofeedback and the group that did not? This should be described and compared in the GA and GB groups with all 48 and 75 participants, not separately for responders and non-responders.

This information is added in table 1:

it is added as supplementary information that compares the concomitant diseases between BFB + versus BFB-.

This paragraph is added in line 208: In the BFB + group, there was more hypothyroidism, more than two concomitant diseases, more concomitant treatments with benzodiazepines, topical estrogens, third-stage analgesics, endovesical treatments with glycosaminoglycans, toxic habits, and history of dystocic deliveries, compared to the groupBFB-; while in the BFB- group there were more concomitant treatments with anticholinergics, a history of hysterectomy and euthotic deliveries (table 1).

4.-Similarly, did patients follow the prescribed regimens equally well in both groups? This should be reported in the results.

The patients received the same follow-up in the two groups. The patients similarly complied with the prescribed treatment regimen in both groups, with the exception of the loss to follow-up of 2 patients in GA, who were included in the failure outcome. This is expressed in rows 218-219. We add an explanatory paragraph.

5.-The authors describe capacity limitations as justification of the imbalanced sample sizes in each arm. How did this vary by study site?

We add this paragraph in the line 220:

They were treated in the Hospital Universitario Nuestra Señora de Sonsoles (05004, Ávila, Spain): GA success: n=3; GA failure: n=1; GB success: n=3; GB failure: n=1; en el Institute of Applied Technology (Physiotherapy Department), Abu Dhabi. United Arab Emirates: GA success: n=3; GA failure: n=1; GB success: n=3; GB failure: n=1; en el Hospital Universitario de Salamanca (37.007 Salamanca, Spain): GA success: n=11; GA failure: n=3; GB success: n=14; GB failure: n=10; Hospital Universitario Virgen de la Vega (37.007 Salamanca, Spain): GA success: n=11; GA failure: n=3; GB success: n=11; GB failure: n=10;

 y en el Hospital Universitario Virgen del Castañar (37.700 Béjar, Salamanca, Spain)

GA success: n=8; GA failure: n=4 pero se perdieron 2 para el seguimiento; GB success: n=13; GB failure: n=9.

6.-How was disease-free time reported? Is this reported by the patient directly to the research team, or does it come by way of the physician during an in-clinic visit?  Related, what are the possible values that it can take?  If at an office visit, are there prespecified times at which it can be reported? Is there any possibility of ascertainment bias between the two groups when measuring this?

We add this paragraph on line 117:

All patients received on paper at the first follow-up daily visit: a card per month, where the days were expressed in squares. If the patient had incidents, she recorded them at the time of suffering them. Likewise, if he was without symptoms, he also wrote it down at the end of the month. Thus, all the patients delivered the evolutionary diaries in the subsequent period. This was done in all patients in both groups. This procedure tries to avoid forgetting.

7.-The definitions of success and failure or not exclusive. Someone who goes from 40 to 70 satisfies both criteria, as does someone who goes from 75 to 85. The authors need an ‘or’ statement to define success, and an ‘and’ statement to define failure. If this interpretation is correct, the authors can simply state that a success is defined as increase of 30 or more points from

The reviewer is right: we add "or" in the definition of success, and "and" in the definition of failure.

8.-The statistical analysis section is terribly incomplete. The authors need to list WHY each type of analysis was done in the context of the study goals. They should also describe a bit more of the detail about how each of the analyses were done. For the multivariate analysis, what were the contributing variables? What was the goal of this analysis? Which analysis was done to examine the treatment effect of biofeedback? Which variables did this analysis adjust for? There’s a lot of output presented without any rationale or explanation.

Following the recommendation number 14 of the reviewer one, the multivariate analysis by the bietapic claster technique was eliminated.

9.-Given the longitudinal at 3, 6, and 12 months, why not use an approach such as mixed effects models or GEE that would allow outcomes to be compared using each time points separately? The authors are losing information about the differences in trends by taking the mean over outcomes within individual. It would be of interest to see plotted the proportion of responders in each arm at each of the 3 follow up time points and then to report on whether the differences between arms are constant in time or increasing or decreasing.

The reviewer is right that it might seem interesting to analyze the results of the SF-36 questionnaire at each control point (3-6-12 months) after finishing the treatment.

We substitute Table 2 of the SF-36 results for the following one, which contains the values ​​and comparisons of the SF-36 questionnaire at the different control points:

SF-36

Group

Check Point

Media

SD

Median

Range

p

BFB+

Pre-treatment

42.98

2.44

42

41-47

Post-treatment 3 months

78.27

13.02

85

51-92

0.0092

Post-treatment 6 months

78.31

13.40

85

49-92

Post-treatment 12 months

75.72

12.81

81

47-90

Post-treatment average

77.29

13.20

84

51.89

BFB+ success

Pre-treatment

42.81

2.66

41

41-47

Post-treatment 3 months

85.38

3.51

86

77-92

0.2284

Post-treatment 6 months

85.52

4.16

86

76-92

Post-treatment 12 months

82.08

3.84

81

73-90

Post-treatment average

84.55

3.15

85

78-89

BFB+ failure

Pre-treatment

43.50

1.57

43.5

42-45

Post-treatment 3 months

56.92

4.68

56.5

51-62

0.8499

Post-treatment 6 months

56.67

5.50

57

49-62

Post-treatment 12 months

52.80

4.21

53.5

47-58

Post-treatment average

55.50

4.7

55.50

51-60

BFB-

Pre-treatment

47.96

6.17

45

40-58

Post-treatment 3 months

73.92

15.97

83

48.95

0.0375

Post-treatment 6 months

72.25

15.69

81

48.90

Post-treatment 12 months

69.76

16.38

80

46-87

Post-treatment average

71.79

15.79

82

50-82

BFB- success

Pre-treatment

48.20

5.78

45

40-58

Post-treatment 3 months

86.90

3.58

87

80-95

0.1028

Post-treatment 6 months

85

3.23

85

78-90

Post-treatment 12 months

83.27

2.43

83

78-87

Post-treatment average

84.77

2.44

85

80-88

3.45B55F48-61B- failure

Pre-treatment

47.61

6.75

45

41-58

Post-treatment 3 months

55.48

3.45

55

48-61

0.2185

Post-treatment 6 months

54.16

3.91

53

48-61

Post-treatment 12 months

50.58

2.40

50

46-55

Post-treatment average

53.35

2.85

52

50-59

SF-36: value in the answers to the quality of life questionnaire SF-36. SD: standart deviation.

10.-The baseline characteristics table (Table 1) should include baseline SF36 and concurrent conditions. These should be reported in each group without consideration of responder status.

Table 1 is expanded by adding the SF-36 value and the concomitant conditions in BFB + and BFB-.

11.-What was the completeness of data in each arm? How many participants provided at 1, 2, or 3 of the assessments at 3, 6, and 12 months?

This comment is added to the results:

Except for two patients in the BFB + failure group who did not carry out the control at 12 months post-treatment, the rest of the patients completed all the controls.

12.-It looks like the authors did a within group (GA and GB) analysis comparing those who were successes and those who were failures to infer treatment effects. This is not the right analysis to determine the effect of BFB, and as such, table 2 should be removed.  The authors need to do a between group analysis that compares the probability of success among those who were in the GA group vs those who were in the GB group.  Unadjusted and adjusted analyses should be presented.  It looks like n(%) responding is reported in lines 210-211 of p. 5, but there is no p-value to assess whether 75% vs 58% is statistically significant.  The authors should further report a measure of effect such as a risk difference, risk ratio, or odds ratio with respective 95% confidence interval.    The primary analysis, however, should be a between groups comparison adjusted for potential confounders, necessarily including baseline SF-36. This can be accomplished through an appropriate regression model.  A measure of effect that considers the difference in % success, adjusted for potential confounders (by definition pre-treatment), along with respective 95% CI and p-value, should be reported as the primary analysis, and differences in mean change from initial SF36, treating response as a continuous variable, should be reported as secondary. The analysis presented in table 4 is close, but it should contain treatment (BFB or not ie GA or GB) as effect, with the treatment difference determined by the coefficient of GA or GB, and post-treatment SF-36 and any other post-treatment variables (time disease free?) should not be in the model.

13.-More on table 4. The logistic model appears very poorly specified. The Wald statistics and standard errors are implausibly huge. The intercept (constant) is nonsensical. This is likely b/c post-treatment SF-36 is in the model and by definition constructs the outcome. 

According to the comments of the reviewer, the expression of the statistical analysis is changed.

14.-The multivariate clustering analysis should be removed. The purpose of this analysis is not stated, and the authors in appropriately combine pre- and post-treatment variables to determine factors that are predictive of success. The results are not clearly presented, and it’s not clear how the authors intend for the analysis to be used. Are they trying to determine a group likely to respond to treatment? Are they trying to assess mechanism?  Moreover, it does not make sense to include post-treatment SF-36 in the clustering since it defines the outcome. This is acknowledged in the discussion, but the authors’ justification does not hold.

The multivariate two-stage cluster analysis is eliminated from the results, therefore also the comments in the discussion.

15.-Figures 1 and 3 are unintelligible. Why are upper and lower Cis presented but not point estimates? And why are there lines connecting points that represent different variables. These are not scatter plots as they are labeled.

According to the comments of the reviewer, we change this figures.

16.-How is ‘influence’ defined in Figure 2?

According to the comments of the reviewer, we change this figure.

17.-10 lines 321-331 contain conclusions based on inappropriate analyses. These statements are not supported. The prognostic value of initial SF36 cannot be inferred when post-treatment SF36 is included since the impact of initial SF36 is likely mediated by posttreatment SF36. The multivariate analysis does not provide evidence of the significance of the difference in proportion of responders.  This should be done by logistic regression or Fisher’s exact test assuming 1 response per participant obtained by average longitudinal response.  The generalized linear mixed model approach mentioned above is an alternative approach that can accommodate longitudinal responses for each participant.

We consider that with the changes made in the presentation of the results, the conclusions are supported.

We appreciate the comment.

Additional issues:

18.-The authors statement of their goal sounds biased in the direction of a positive result. They should state that they want to ‘evaluate’ or ‘determine’ the effect of biofeedback, not to ‘prove’ it.

The reviewer is right: we change the objective statement: it is stated like this:

To evaluate the effect of pelvic floor muscle training with BFB on HRQOL as a complementary treatment in patients with BPS/IC.

19.-The group names GA and GB are not intuitive. Why not label the groups as something that reflects the therapy, like BFB+ and BFB-?

We change it. Thank you.

20.-On p. 3 line 96 it should be ‘BRB on HRQOL’ rather than ‘BRB in HRQOL’.

We change it. Thank you.

Reviewer 2 Report

Abstract

Methods

Provide full meanings of GA, GB and other acronyms when first mentioned

Results

Were the values for mean age and mean BMI the same in both groups? Better to report the value of each variable separately for each treatment group and then report whether any observed differences in variables are significant. Only important outcomes should be reported in the abstract e.g. health-related quality of life

Main manuscript

Introduction

Authors should be systematic in their discussions: first discuss the interventions (biofeedback, pelvic floor muscle training (PFMT)), followed by quality of life and health indicators.

Authors should briefly describe the mechanism of action of PFMT in the management of BPS/IC

Methods

Lines 100/101: what do the authors mean by this statement: ‘Inclusion was prospective, consecutive, and exhaustive, between December 2013 and December 2016’?

Sample selection: could the authors provide a brief explanation of the significance of ESSIC 2C-3C for readers who may not understand what it means?

Procedure: should a copy of the SF-36 HRQOL questionnaire be provided as an appendix?

Statistical analysis: student’s t-test, chi-square test, ANOVA, etc, were used to analyze what and what? Could the authors specify?

Results

As mentioned earlier, authors should report the value for each variable by treatment group to allow for meaningful comparison between the treatment groups. In the alternative, authors can refer to the table presented in the  text for details of the findings rather than report the values for the whole treatment sample. Readers are more likely to focus on the values for each treatment group.

At what points were the variables (outcomes) assessed? Immediately after treatment? At 3, 6, or 12 months after treatment? Did the observed differences in treatment outcomes persist long term? Answering these questions will have important bearing on the conclusions from this study

Any limitations of the study?

Author Response

REVIEWER 2. ANSWER IN RED AND ADD THE MODIFICATIONS TO THE MANUSCRIPT.

Abstract

Methods

1.-Provide full meanings of GA, GB and other acronyms when first mentioned

Answer:

The acronym GA and GB have been changed to more intuitive BFB + and BFB- that are defined the first time they are used.

Results

2.-Were the values for mean age and mean BMI the same in both groups? Better to report the value of each variable separately for each treatment group and then report whether any observed differences in variables are significant. Only important outcomes should be reported in the abstract e.g. health-related quality of life

Table 1 shows that the age was similar in both groups (p = 0.0535) but the BMI was higher in BFB- (p = 0.04463). There was a misprint in the abstract that we corrected, specifying that the BMI was different between groups.

Main manuscript

Introduction

3.-Authors should be systematic in their discussions: first discuss the interventions (biofeedback, pelvic floor muscle training (PFMT)), followed by quality of life and health indicators.

Following the recommendation of the reviewer, we changed the order of the introduction paragraphs, putting the contents of lines 68-79 after lines 80-94.

4.-Authors should briefly describe the mechanism of action of PFMT in the management of BPS/IC

This paragraph is added in the introduction:The objective of biofeedback of pelvic floor is to reestablish a balance in the physiological functioning of the pelvic structures, through regular training that breaks the stress that the various pathologies can cause in the pelvic floor, including long-term painful conditions, which they will cause reflex contractures. Biofeedback acts retroactively from the reflex effector organs towards the nervous structures. It aims to regulate reflections in an inverse way. This has already been demonstrated in previous investigations of our research group (B. Padilla-Fernandez; A. Gómez-García; M.N. Hernández-Alonso; M.B. García-Cenador; J.A. Mirón-Canelo; A. Geanini-Yagüez; J.M. Silva-Abuín and M.F. Lorenzo-Gómez. Biofeedback with Pelvic Floor Electromyography as Complementary Treatment in Chronic Disorders of the Inferior Urinary Tract. In Título del libro: Electro diagnosis in New Frontiers of Clinical Research. Editor Hande Turker© . Rijeca (Croatia) Edit InTech, 2013. I.S.B.N. 978-953-51-1118-4. Cap. 14 PP. 287-311).

Methods

5.-Lines 100/101: what do the authors mean by this statement: ‘Inclusion was prospective, consecutive, and exhaustive, between December 2013 and December 2016’?

We change it to:

Inclusionwas randomized by each doctor prospective between December 2013 and December 2016.

6.-Sample selection: could the authors provide a brief explanation of the significance of ESSIC 2C-3C for readers who may not understand what it means?

We change this phrase:

For inclusion, signs of cystitis on cystoscopy, and inflammation findings in the bladder biopsy were needed  (ESSIC 2C-3C)

To:

For inclusion, signs of cystitis on cystoscopy, and inflammation findings in the bladder biopsy were needed: types of lesions 2C and 3C of the European Society for the Study of Interstitial Cystitis were included: that is, presence of glomerulations (2C) or Hunner's lesion (3C)

7.-Procedure: should a copy of the SF-36 HRQOL questionnaire be provided as an appendix?

The SF-36 questionnaire used is attached as supplementary material.

8.-Statistical analysis: student’s t-test, chi-square test, ANOVA, etc, were used to analyze what and what? Could the authors specify?

Descriptive analysis, Student's test to compare continuous variables, chi square and ANOVA to perform the analysis of variance and logistic regression analysis that includes multivariate analysis were used. That is, routine analysis has been used in medical scientific studies.

We appreciate the comment.

Results

9.-As mentioned earlier, authors should report the value for each variable by treatment group to allow for meaningful comparison between the treatment groups. In the alternative, authors can refer to the table presented in the  text for details of the findings rather than report the values for the whole treatment sample. Readers are more likely to focus on the values for each treatment group.

The expression of the statistical analysis is changed, although the results and the meaning of the results are the same.

10.-At what points were the variables (outcomes) assessed? Immediately after treatment? At 3, 6, or 12 months after treatment? Did the observed differences in treatment outcomes persist long term? Answering these questions will have important bearing on the conclusions from this study

In both methods and results, it is expressed that the controls are carried out at 3, 6 and 12 months after the end of the treatment. The information on the results has been expanded in the different control points.

11.-Any limitations of the study?

This paragraph is added in the discussion:

The limitations of this study are those inherent to the interstitial cystopathy disease itself. All patients were diagnosed following the same diagnostic protocol. The two main types of interstitial cystopathy were taken into account, as specified in Methods: the definition of lesions in the cystoscopy of the European Society for the Study of Interstitial Cystitis has been used for the study of C2 and C3 interstitial cystopathy (3). Despite this, it is known that there is variability in the disease between different patients, since it is difficult in this disease to identify precise stages.

On the other hand, a very strong point is the long experience of the research team in the management of the disease and the coherence and rigor of the follow-up protocols in all patients (M.F. Lorenzo Gómez. Título: Cistopatía Intersticial. ISBN 84-95670-63-1. Depósito Legal: B. 43.776.2003.

Editores Montserrat Espuña; Jesús Salinas©. Editorial Ars Médica. Barcelona, España 2003.

Tipo de producción: capítulo de libro. Cap. 22 PP. .305-316

Título del libro: Tratado de Uro ginecología. Incontinencia Urinaria. Barcelona. Tipo de soporte: Libro en papel pasta dura.

)(Título del trabajo: Efecto del Tratamiento con Anti-inflamatorios NoEsteroideos en el Síndrome de Dolor Vesical/ CistopatiaIntersticial.

Autor/es: Marta Raquel Lopes Fernandes Vale-Matos, Paula Mendez Garcia, Monica Sanchez Jaen, Monica Paola Cordeque Mejia, Teresa Hernandez Sanchez, María Helena Garcia Sanchez, María Begoña Garcia Cenador, María Fernanda Lorenzo Gomez.

Tipo de participación: comunicación oral Tipo evento: Congreso internacional. Nombre del Congreso: XV Congreso SINUG 2018 Ciudad de celebración: Sevilla Fecha de celebración: 15/11/2018 Fecha de finalización: 17/11/2018

Entidad                           organizadora:        Sociedad      Iberoamericana         de     Neurourología       yUroginecología

)(27.-Nombre de la actividad: Eficacia de la neuromodulación como tratamiento de la vejiga dolorosa o cistopatía intersticial en varones y mujeres.

Autores:BarrosoLópezK.R.,BliekBuenoK.,CanteroAcedoA.,DomínguezGarcíaJ. Maria Fernanda Lorenzo Gomez.ComunicaciónOral:XIIICongresoNacionalde Investigación de Grado en CC de laSalud

Entidad organizadora: Universidad de Alcalá. Madrid.

Título de la reunión: XIII CONGRESO NACIONAL DE INVESTIGACIÓN DE GRADO EN CC DE LA SALUD

Lugar: Madrid. Fecha: 30/03/2017 - 01/04/2017.

).(MF 2004, Jiménez, Gómez et al. 2009).

Round 2

Reviewer 1 Report

The authors are still presenting a lot of inappropriate outcome-stratified data analyses (eg association models for successful vs failed treatment), and they were not responsive to the recommendation about what the primary analysis should be (a multivariable model with BFB+ vs BFB- as a predictor in the model). There are also a lot of unnecessary associational analyses relating variables to response or failure that don't inform the treatment effect of interest. Finally, the revised table 1 analyses show that there were many differences between the two arms despite the randomization, especially in conditions and treatments, so it's not possible to rule out that confounding or different intermediate therapy is explaining the difference in the probability of response between the BRB+ and BFB- groups (which they still do not provide an unadjusted P value value or measure of effect and 95% CI for; they should include a RR or RD with p-value and CI for 76% vs 58% response in BFB+ vs BFB-). Given the large number of variables that they'd need to adjust for in a small sample, they could consider a propensity score adjusted model for the probability of success given treatment (BFB+ vs BFB-) and the estimated propensity score for BFB+ given pretreatment confounding variables.  

Unfortunately, as it currently stands, the manuscript still doesn't use correct methods.  They really need someone with strong epidemiology training to provide input.

Author Response

Following the reviewer's instructions, any comparison between the success and failure subgroups is eliminated.

The relative risk analysis is added, with the 95%, 76% and 58% confidence intervals as requested by the reviewer, that the patients go to the BFB + group or the BFB - group.

It is observed that the presentation of the results as requested by the reviewer, does not alter the conclusions, but even makes the success results more powerful in the group of patients treated with BFB, since it is observed that they presented worse conditions, more diseases concomitants.

Thank you very much.
